# Challenges to the Application of Integrated, Personalized Care for Patients with COPD—A Vision for the Role of Clinical Information

**DOI:** 10.3390/jcm9051311

**Published:** 2020-05-02

**Authors:** Lowie E.G.W. Vanfleteren, Alex J. van ‘t Hul, Katarzyna Kulbacka-Ortiz, Anders Andersson, Anders Ullman, Martin Ingvar

**Affiliations:** 1COPD Center, Department of Respiratory Medicine and Allergology, Sahlgrenska University Hospital, 413 45 Gothenburg, Sweden; katarzyna.kulbacka-ortiz@vgregion.se (K.K.-O.); anders.andersson@lungall.gu.se (A.A.); anders.ullman@vgregion.se (A.U.); 2Department of Internal Medicine and Clinical Nutrition, Institute of Medicine, Sahlgrenska Academy, University of Gothenburg, 413 45 Gothenburg, Sweden; 3Pulmonary Diseases, Radboud University Medical Centre, 6500 HB Nijmegen, The Netherlands; Alex.vantHul@radboudumc.nl; 4Osher Center for Integrative Medicine, Karolinska Institutet Stockholm, 171 77 Stockholm, Sweden; martiningvar@me.com

**Keywords:** COPD, person-centered care, clinical health informatics, care plan, chronic disease, multimorbidity

## Abstract

Chronic Obstructive Pulmonary Disease (COPD) is a complex disease defined by airflow limitation and characterized by a spectrum of treatable and untreatable pulmonary and extra-pulmonary disease characteristics. Nonpharmacological management related to physical activity, physical capacity, body composition, breathing and energy-saving techniques, coping strategies, and self-management is as important as its pharmacological management. Most patients with COPD carry other chronic diagnoses and this poses a key challenge, as it lowers the quality of life, increases mortality, and impacts healthcare consumption. A personalized, multi-, and interprofessional approach is key. Today, healthcare is poorly organized to meet this complexity with the isolation between care levels, logic silos of the different healthcare professions, and lack of continuity of care along the patient’s journey with the healthcare system. In order to meet the criteria for integrated, personalized care for COPD, the structural capabilities of healthcare to support a comprehensive approach and continuity of care needs improvement. COPD is preeminently a disease that requires a transition from a reactive single-specialty approach to a proactive interprofessional approach. In this study, we discuss the issues that need to be addressed when moving from current health care practice to a person-centered model where the care processes and information are aligned to the individual personal needs of the patient.

## 1. Introduction

Chronic Obstructive Pulmonary Disease (COPD) is a complex disease defined by persistent airflow limitation and characterized by a spectrum of treatable and untreatable pulmonary and extra-pulmonary disease manifestations [1,2,3]. Nonpharmacological management related to physical activity, physical capacity, body composition, breathing and energy-saving techniques, coping strategies, and self-management is as important as its pharmacological management [2,4]. Most patients with COPD carry other chronic diagnoses and this poses a key challenge, as it lowers the quality of life, increases mortality, and, importantly, impacts healthcare consumption [5,6,7]. Addressing this complexity is a true challenge, as healthcare has been organized from a reactive unidimensional, biomedical, and single-organ specialty approach [3].

The concept of personalized medicine (PM) in chronic care focuses on the ability to continuously adapt all health care efforts along the health trajectory to cure/meet the symptoms and support health and individual autonomy [8,9]. PM reflects the optimism of improved care by focusing on all treatable traits, well-being and autonomy of the individual, and quality of life, which can be improved even in severe chronic disease [10]. PM combines knowledge from different domains (e.g., behavior, social domain clinical observations, clinical chemistry, medical imaging, cellular analysis, proteomics, and genetics) in order to provide a comprehensive basis for the individually tailored treatment plan for intervention and advice that serves the patient. However, at the same time, the quest for personalization drives complexity in healthcare, as “one-size fits all” care programs cannot be applied. The maintenance of continuity of knowledge, decisions, and recommendations becomes a major challenge since COPD is a progressive chronic disease often accompanied by several co-diagnoses.

In this article, we discuss the concept of PM in COPD where patients have different needs and commonly carry more than one diagnosis. Whereas not all promises of PM may be realized in COPD, we demonstrate a model that allows focus on individual treatable traits, secondary prevention, improvement of quality of life, patient autonomy, and compliance to clinical guidelines/recommendations [11].

Standard care pathways (care plans) have been suggested as a means to reduce variability in healthcare practice [12]. There is a fine line between the acceptable standardization of care and an unwanted constraint of the clinical decisions and use of clinical experience. However, the present practice of clinical care yields a very high variability between practitioners and clinical decisions that should not vary, i.e., deviance from guidelines, and, consequently, low-quality care for the individual [13].

## 2. The Clinical Needs in COPD

COPD is defined by the Global Initiative for Chronic Obstructive Lung Disease (GOLD) as “a common, preventable and treatable disease characterized by persistent respiratory symptoms and chronic airflow limitation due to airway and/or alveolar abnormalities, usually caused by significant exposure to noxious particles or gases and influenced by host factors including abnormal lung function development.” Significant comorbidities may have an impact on morbidity and mortality [14]. COPD is a highly prevalent disease, and projections for the future suggest a further rise in the prevalence of COPD patients, especially of these with severe or very severe stage [15]. COPD often puts a massive burden on those affected, even when they are in a relatively stable phase of their disease or only have mild to moderate airway obstruction [16]. A recent cross-sectional survey in the U.S. and five Western European countries revealed that, despite using appropriate standard-of-care medications, patients with COPD still have a significant impaired health status [17]. Moreover, the impact of COPD places an inordinate burden on healthcare resources given the significant direct and indirect costs of care [18]. The large majority of these costs are related to comorbid disease, which is recognized in the definition of COPD. Indeed, a study performed in Sweden considering real-life evaluations of patients with COPD showed that direct costs were driven by non-COPD-related hospital nights [6]. In this context, COPD has been put forward as the pulmonary component of multimorbidity, and COPD has been translated as Comorbidity with Pulmonary Disease [19]. Indeed, patients with COPD almost invariably carry other diagnoses, thereby increasing the complexity of the clinical decisions [2].

Exacerbations, or sudden flare-ups of the disease, may be present with a major impact on the clinical course of the disease [20,21]. The presence, intensity, and reoccurrence of exacerbations is highly variable between and within patients with COPD. The large unpredictability of exacerbations, as well as the heterogeneity in cause, pose a significant problem. Indeed, exacerbations can be related to a flare-up of eosinophilic inflammation in a proportion of COPD patients, but can also be related to viral or bacterial infections, or to deconditioning, decontrolled breathing patterns, and dynamic hyperinflation, as seen in association with anxiety or panic attacks or comorbid conditions (e.g., cardiovascular disease, pulmonary embolism, etc.) [22,23,24]. The acute exacerbations and the fear of exacerbations pose a significant problem for the individual patient and also cause potentially unnecessary health care consumption [25].

COPD is considered a complex, heterogeneous disease [26], diagnosed through the presence of persistent airflow limitation and characterized by a spectrum of other treatable and untreatable pulmonary and extra-pulmonary disease manifestations that require complex management. The combination of both complexity and heterogeneity of the disease leads to poorly predictable treatment responses [10]. Therefore, a multidimensional patient profiling is crucial to identify the right COPD patient for the right treatment. Dynamic, personalized, and holistic approaches are needed to tackle this multifaceted disease and to ensure personalized medicine [26].

In chronic complex care, it is of importance to organize care according to the biopsychosocial model of disease [27]. This model is a 40-year-old concept, but few tools of support for the model have been available in healthcare. In COPD with co-diagnoses, the focus is often set only on the airway symptoms (biological factor), and healthcare often fails to fully address other important dimensions of the disease, such as socioeconomical support, psychological status, and measures for secondary prevention. It is commonly reported that patients continue to have symptoms after optimal pharmacological treatment, that one-third of patients on triple inhaler therapy continue to have exacerbations, and that one-third of the patients continue to smoke in advanced stages [28].

The WHO definition of “health” reads “a state of complete physical, mental and social well-being and not merely the absence of disease or infirmity.” When this definition was coined in 1948, infectious diseases were the main problem. Today, however, chronic and lifestyle diseases are much more prevalent, particularly in the Western world, and 95% of the present healthcare budget is spent on medication and intervention in spite of early understanding that prevention is a core task in medicine [29]. Newer definitions of health are more adaptable to COPD care. Huber and van Vliet recently defined health as “the ability to adapt and self-manage, in light of the physical, emotional and social challenges of life” [30]. Health is no longer considered as a static condition, but rather as the dynamic ability to adapt and to manage one’s own well-being. The way in which “health” is defined has a significant influence on how we organize and use healthcare. Research by Huber and Van Vliet has shown that patients consider these abilities very relevant. By shifting the emphasis to resilience and well-being (rather than ill-health), the new health concept helps policymakers and politicians change their thinking about healthcare and disease prevention. This change is urgently needed if we want to maintain high-quality care that is also affordable. Thus, proper personalized medicine entails addressing multimorbidity, well-being, social issues, and secondary prevention. Hence, each healthcare professional should share this broad model of care and include the patient in the decision-making and planning.

The application of personalized medicine (PM) in COPD needs to address dimensions such as decreased quality of life, shortened life expectancy, and increased healthcare consumption. A key objective should be stratification of treatment to meet the challenge via addressing of identifiable treatable traits and secondary prevention strategies, i.e., a number of dimensions that are not fully independent of one another [2,31,32]. The multidimensionality suggests a need for personalized care along the care trajectory, and this need is confirmed by the fact that the step-therapy approach, in which a linear progressive model for COPD is applied, has not been successful. Individual variability in both clinical presentation and treatment response is increasingly recognized. Even GOLD has removed the severity of airflow obstruction from its treatment decision algorithms, which is now defined based on symptoms and exacerbation frequency on the one hand, and biomarkers and (un)response to treatment on the other hand [14].

## 3. Capitalizing on Patient Engagement

COPD is a chronic condition in which correct spirometry and flow-volume measurements base the proper diagnosis and classification of the severity of the airway obstruction. However, these measures poorly reflect the individual burden of disease and quality of life and therefore provide an insufficient basis for personalized care [26]. As stated, multimorbidity is the rule rather than the exception in patients with COPD [7], and exacerbations are largely unpredictable events that have a tremendous impact on the individual patient [33]. In addition to the complicating physical factors, behavioral and psychosocial aspects are well recognized and modifiable factors to affect the impact of the disease on patient’s lives [34]. This realization of the complexity in the disease trajectory leads to the need for an integrated, personal approach in the care of patients with COPD [31]. A COPD care strategy should entail measures to continuously assess the burden of disease (including disease severity, disease activity (symptoms, exacerbations), proposed as a “COPD dashboard” by Agusti et al. [8], but psychological status, social support network, and quality of life should also be taken into account. Strategies for early detection and prompt intervention and to decrease the risks of exacerbations are also important. In the advanced stages, this includes measures of patient education for self-management strategies [3,35,36] and the regular (remote) collection of patient-reported outcome measures (PROMS) in combination with prepared routes for the patients to reach primary/secondary care in ensuing acute exacerbation of COPD (AECOPD) (Table 1). Modern technology provides a unique opportunity here. For example, secure mobile phone- or tablet-based data communication allow the patient to deliver necessary information for continuous decision-making [37]. The challenge here is not the ability to communicate, but rather the ability to handle the information rationally [38] and meet the patient’s interest in the decisions and planning. There are a number of dimensions in the treatment that may be improved by patient involvement. Patient-reported data are of importance when performing healthcare quality assessments. Compliance to treatment and measures of secondary prevention are supported when the patient is a collaborating agent with access to patient education and the ability to record, e.g., physical activity and smoking cessation efforts.

## 4. Continuity of Care

Table 1 points to the need for three general reforms of healthcare. First, interprofessional sharing of decisions and knowledge is needed to serve the patient with a consolidated and evidence-based diagnosis and treatment. Second, in order to maintain continuity of care content along the care trajectory, information needs to be shared along the patient’s care trajectory, especially when the patient moves to the next provider. Thus, not only the health records from electronic medical record (EMR) (looking back perspective) need to be accessible, but also the multi-professional care plan (planning ahead perspective), as well as planned data capture from care. Additionally, the patient must have the ability to communicate. Third, a patient-centric model should provide the patient with continuous means to communicate with healthcare, report health outcomes, and receive support for health literacy that is individually adjusted. Hence, we foresee a rapid need-driven development of the information support in healthcare. In personalized medicine, the patient is the central resource of information on his/her own life and disease trajectory [3]. A large part of the COPD care for patients can be structured and thereby organized to ease the burden of administration (PROM collection, health literacy education, scheduling and rescheduling) and limited (asynchronous chatbots, not mixing acute contacts with general contacts channels). Above all, proper information reduction before presentation to healthcare and patients decreases the burden.

When the complexity and heterogeneity of patients with COPD is incorporated in regular care, it invokes a paradigmatic shift in which COPD care is transformed from a biomedical reductionist approach, i.e., mainly directed at the airway obstruction, which can be provided by a single healthcare professional. Instead, a holistic model emerges with a multidimensional biopsychosocial approach in both assessment and the (chronic) management, which requires teamwork in the involvement of multiple and different healthcare professionals (HCPs). Many and important treatable traits in patients with COPD are best met with non-pharmacological interventions delivered monodisciplinary by a dietician, physiotherapist, occupational therapist, psychologist, social worker and specialist nurse, together the minimum recommended personnel for the multidisciplinary team and necessary to address the complexities associated with the disease [2,39,40] (Figure 1). Engaging a multidisciplinary team enables the management of the complexities that characterize this patient population and could be a useful platform to identify treatable traits and to implement a targeted treatment program based on these traits. Such a program has been shown to reduce hospital admissions, improve health status, reduce exacerbations, and reduce the number of bed days [41]. Referral pathways to other specialties for the treatment of common comorbidities or traits should also form part of a multidisciplinary team protocol. In such context, patients can be intensively investigated, and diagnostic equipment, multiple treatment options, and multidisciplinary expertise are available. Indeed, a treatable traits optimal management approach for patients with COPD requires a multidimensional assessment and targeted treatments [39,40]. The current healthcare model where patients move between care levels, care providers, and professions along the disease trajectory does not suffice. Variability in decisions, discontinuity in planning and documentation, and payment models that conserve old care patterns contribute to a varying quality for this type of care. To meet this identified weakness, the development and implementation of integrated care healthcare networks are advocated and have indeed been found to improve some aspects of the continuity of care [42]. Such care models are expected by the patients. Twenty years into the internet era, today’s patients are used to seamless and easy access services in all corners of life.

Many decisions that are made in the clinic with or on behalf of the patient are sometimes taken without an explicit basis for the decision. It is actually a minority of decisions and recommendations/guidelines that have a top-level evidence base [43]. The sharing of reasons for decisions is required for true interprofessional work models and especially if modern patient involvement is sought. The definition of treatable traits, goals for intervention, and common assessments of outcomes provide a means for the sharing of knowledge and systematic documentation along the disease trajectory. The GOLD classification of COPD severity is a good start, as it carries unified information to all professions about core aspects of COPD, including the severity of airflow limitation (severity), disease impact measured by validated symptom questionnaires (COPD assessment test (CAT and/or modified Medical Research Council (mMRC)), dyspnea score (impact), and a measure of exacerbation frequency (activity of the disease).

The agreed measures for continuous classification of impact of disease is embedded in the interprofessional team. An interprofessional understanding of the concept of COPD disease characterization is important for the coordination of intensity of interventions from different healthcare professionals. The steps toward secondary prevention and general lifestyle support vary markedly across the different stages. The interprofessionally shared understanding of the current disease state is essential as to maintain all advice adjusted to the disease stage and the patient individual traits and preferences.

## 5. Complex Chronic Disease and Learning Healthcare

In chronic disease, and specifically in multimorbidity, a patient’s healthcare trajectory is complex and entails frequent patient visits to many corners of healthcare. Thus, it is a challenge to share the relevant knowledge across all these different instances.

A shared access to a common electronic medical record (EMR) is only a partial solution, as the EMR holds sparse and unstructured information to lead the common plans for the patient. A modern solution should entail the sharing of personalized care pathways where the intention of the team effort to improve the health status of the patient is operationalized (Figure 2). Historically, the medical health record was constructed to create a record of events and thereby legally and economically protect healthcare professionals and provide billing support to the health organization. This organization-centric rather than patient-centric view has delayed the developments of the aspects of patient service and logistics support [44]. An interprofessional work model enforces the construction of a united work process, and the health information tools should master the real-time sharing of information of not only the historical health record, but also recommendations and care planning. Many efforts are ongoing to develop such a change, but, so far, there are few clinically available solutions. A modern solution should be based on structured information with underlying terminologies that allow for continuous aggregation of data that are necessary for the optimization of treatments. An important byproduct of using a clinical pathway is that the capture of group level data can be planned at the same time. Thereby, local experience may be gathered, and the concept of learning healthcare may become institutionalized.

## 6. Challenging the Healthcare Information Silos

The patient with advanced COPD is typically managed in both primary and secondary care, and, in secondary care, the patient is managed in both inpatient and outpatient care. The patient relies on inpatient care for the management of severe exacerbations and on primary care for management of unscheduled non-severe exacerbations. The multimorbid COPD patient have multiple caregivers that require coordination. As the clinical burden of COPD is heavily related to comorbidities, the capability to handle such complexity in COPD in the trans-professional team setting is of great importance [45]. This represents a specific challenge, as medical insights vary across professions. The focus on conveying the dimensions of quality of life and secondary preventions strategies is a good start, but a clinical conference where the findings and recommendations of different healthcare professionals are amalgamated to a cohesive strategy is most often necessary.

## 7. A Digital Concept: Supporting the Patient Centered Process

IT solutions should have the potential to support and reinforce the deployment of an innovative healthcare model for patients with COPD, which is scalable to include other chronic noncommunicable diseases, as well as information from other healthcare levels and providers. The IT solution needs to support the clinical approach that considers a structured and comprehensive assessment, a patient-tailored integrated care plan, involving the empowered patient and connecting professions and healthcare segments. This care plan and all other patient information should travel with the patient along the care trajectory (Figure 3). This means that primary and secondary care and hospital inpatient care should be connected to the same treatment plans according to the concept of the comprehensive care unit [46]. The current service and IT model in healthcare is most often profession- and care unit-centered. Thus, the documentation supports record-keeping (legal and economic accountability), and minimal solutions exist for support of work procedures. The information handling should be reversed and become patient-centric, and the architecture should include the documentation of the medical process, and, importantly, should also include patient-reported data and support the planning of the trajectory of the patient. The latter information should be shared between healthcare levels, specialties, HCPs and, above all, the patient. Whereas the information is handled in a single system (the middle layer that connects the patient and the connected caregivers), the local record-keeping is done in the EMR and held at each unit for accountability. The care plans travel with the patient. The conceptually elaborate information model in the middle layer allows for an annotation that enriches each data point. For example, a blood pressure data point could contain the information of context, reason and circumstances of the measurement. Such meta-information creates a basis to access determinants for documentation, quality work, and analytics/research. Decisions are supported in that all actors have the same information available and will be guided through the predetermined decision steps regarding care logistics.

## 8. Limitations

Along this manuscript, we highlight some limitations in relation to the multidisciplinary structural assessment and management of the patient, the concept of the patient in charge of his own disease trajectory, the sharing of treatment plans across healthcare professionals, and the role of IT in all these developments. In this paragraph, we aim to summarize these and elaborate on limitations and shortcomings as of today.

Moving away from the traditional single point of decision (the expert model) to the model with shared decision-making in an interprofessional team is a nontrivial step that requires an organizational revolution. The interprofessional model risks adding complexity and thereby adding costs for healthcare. Without a proper adjusted work organization, coordination efforts become a serious burden and lower the efficiency of the healthcare unit. The planning step that involves all professions is a key to find the most efficient local solution for the work.

Although we pin huge hope on IT solutions for communication and coordination between therapist(s) and the patient, it could seem to paradoxically have the danger in itself of atomizing care with no one obviously in charge. However, the IT solutions are thought to build support with the multi-professional organization in order to handle individually tailored support. Without such an upgrade, there is a high risk that, with increasing complexity, each visit of the patient becomes burdened with the need for a detailed analysis and a detailed description of treatments and recommendations. Importantly, the priority of decisions and priority of recommendations must remain unambiguous or confusion may occur. A multi-professional work organization must therefore prioritize planning, including the pathway for decisions, in order to provide a united view. Increased patient engagement and participation in the decision making emphasizes this need for care coordination.

The transition to an interprofessional model with the “autonomous” patient as the driver might be criticized as an ideology. This might indeed be difficult to reach with current EMR technologies. A reformed ICT support includes the ability to plan the standard clinical pathway where decision points are identified, to individualize the pathway according to the patient needs, and to allow real-time access to all planning information for all professionals to enable them to supply information to patients a practical way. Whereas many of the dominating EMR vendors have identified these needs, most of them have a long way to go before they can provide the needed support for the proposed model.

The issue of integrity of sensitive health data must be guarded in the design of the information support. The proposed model for information sharing along the patient trajectory supports the autonomy of the patient and has the potential to improve quality and efficient resource use in healthcare. Therefore, as long as the data that is collected and shared is technically protected, the GDPR and similar legislations do not preclude the information tools that we envision here.

## 9. Conclusions

COPD is a complex and heterogeneous disease from both the patient and from the healthcare organizational side. A patient with COPD is commonly a multimorbid elderly subject that requires a holistic, comprehensive approach with specialized and individualized treatments from different medical and paramedical specialties, as well as a management approach that commonly lies across the classic healthcare silos. The complexity of the causes of the impaired health status, that is, the number and severity of treatable traits, should be prioritized when making choices with regard to which care provider, where and when, and to which patients to provide intervention. Interprofessional and trans-organizational patient-centric care with common treatment plans is the key. The current care information systems, which are site-centric, single health profession-centric, and focused on the documentation of past events and observations, provide a hindrance to the development of person-centered care. Breaking the silos rests on a commitment from professionals, healthcare providers, and payers. Trans-professional work models depend of qualified support from healthcare information systems, and in the absence of qualified support, such models become economically unsustainable and inefficient because of all the verbal information that is needed to sustain individual care plans. This leads to imprecise decision-making and imprecise communication with the patient. With correct information sharing, the goal can be reached of seeing the patient as a primary asset in the multi-professional team that can together create continuity of care decisions and care interventions. In this way, complexities in chronic complex care of COPD can be managed, moving away from a reductionistic airway-centered approach to a holistic, multi-professional personalized, and patient-related outcome approach. In the end, this should lead to an improved quality of care with better adherence to clinical guidelines and recommendations and to an improved quality of life for the patient with reductions in both burden of disease and burden of care.

## Figures and Tables

**Figure 1 jcm-09-01311-f001:**
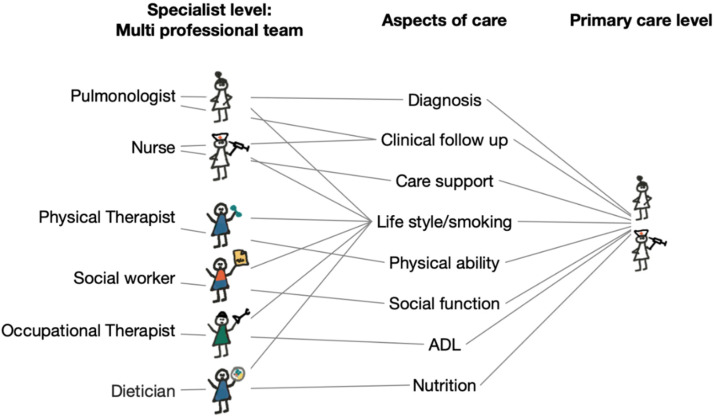
The multi-professional team provides a highly specialized analysis and treatment recommendations for different domains of patient needs. The knowledge pertaining to the patient must not be lost when the patient transfers to primary care where, most often, the care process is confined to the GP and the nurse. Also, the multi-professional team should have tools for a rational sharing of knowledge to motivate the costly model for care organization.

**Figure 2 jcm-09-01311-f002:**
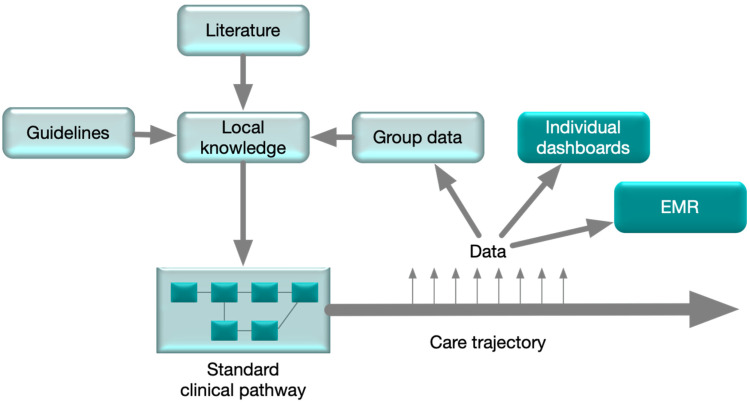
The standard clinical pathway was constructed based on national guidelines and evidence from the literature. During the execution of the plan, data is registered to the EMR and the individual data dashboard that supports the care in the interprofessional group. Data can also be collected on the group level and be used to update the local standard clinical pathway. Patient reported data was included as important potential modifiers of the standard clinical pathway.

**Figure 3 jcm-09-01311-f003:**
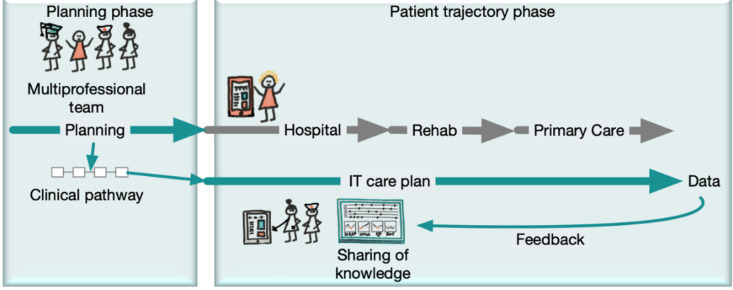
A schematic drawing that demonstrates the organization of the desired IT support system. The multi-professional team designs an optimal clinical pathway and key clinical indicators of disease severity and activity. The clinical pathway is translated to a care plan that provides continuity in the knowledge building along the care trajectory irrespective of provider. The clinical structured data are available in real-time and provide feedback to both healthcare and patients in real-time. Hence, the multi-professional group is supplied with tool for process optimization, patient service, and knowledge sharing.

**Table 1 jcm-09-01311-t001:** Increasing the ambition of continuity, patient involvement and understanding compliance often entail patient education and increased focus on patient-reported outcome measures (PROMs) as a tool.

Traits	Low Ambition Practice	High Ambition Practice
Tobacco smoking	Advice on smoking cessation, possibly a prescription for nicotine replacement, or pharmacological support	Advanced patient education, agreement of strategy and goals, regular follow-up and PROM inclusion
Physical activity	Advice on physical activity	Advanced patient education, agreement on strategy and goals, follow-up including wearables, PROM inclusion
Physical capacity	Advice on exercise training program at home	Exercise training program customized to the specific needs of the patient based on a thorough assessment of exercise limiting mechanism(s)
Activation for self-management	Simple advice and providing generic educational materials	Individualized intervention(s) based on an assessment of individual needs to improve knowledge, skills and self-efficacy for self-management
Weight regulation	Instruction on how to gain, maintain or lose weight	Advanced patient education, agreement on strategy and goals, follow up including wearables, PROM inclusion
Prevention of exacerbations	Individualized pharmacological intervention, vaccination, general information,	Individualized pharmacological intervention, vaccination, advanced patient education, advice on early detection
Symptoms of anxiety and/or depression	General information, possibly a prescription for an anxiolytic	Advanced patient education, cognitive-behavioral therapy if appropriate, PROM inclusion
Pharmacological treatment	All prescribed medications under control, standard drugs	Written, easy-to-understand information, PROM for compliance, side effects, and understanding
Self-management strategies	Instruction to seek help if symptoms are severe	Advanced patient education on early detection. Access to a fast route to specialized care in ensuing AECOPD. Individual care plan with the goal to minimize the risk. Collaboration between care levels in order to provide continuity of care plan.
Co-diagnoses	Instruction to seek medical attention with primary care or other specialties	Full symptom array including assessment and treatment advice, optimization of care, and follow-ups on each. Strong collaboration with primary care as to support the individualized care plan. Inclusion of PROM that also covers treatment success of comorbidities.

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
