# Peer review of "Challenges to the Application of Integrated, Personalized Care for Patients with COPD—A Vision for the Role of Clinical Information"

_jcm, 2020, doi:10.3390/jcm9051311_

Round 1
Reviewer 1 Report
none
Author Response
Thank you.
Reviewer 2 Report
This review paper is important and informative for the personalized care for patients with COPD.
- The figure to explain clinical needs in COPD is helpful for readers to understand. Especially, the figure having part for multidisciplinary approach (for example, nutrition, psychological, comorbidities, COPD symptoms, rehab) is helpful.
- Regarding IT, the authors provided the role of IT as integrated tool for all information. Please provide the figure how IT can help all information integrated or how IT can incorporate with care givers as well as COPD physician and experts with other parts (rehab or psychology..)
Author Response
- The figure to explain clinical needs in COPD is helpful for readers to understand. Especially, the figure having part for multidisciplinary approach (for example, nutrition, psychological, comorbidities, COPD symptoms, rehab) is helpful.
Response:
The authors provided a figure (Figure 1) with the description to illustrate the clinical needs in relation to COPD, specially showing the importance of multiprofessional team and different aspect of patient’s care.
- Regarding IT, the authors provided the role of IT as integrated tool for all information. Please provide the figure how IT can help all information integrated or how IT can incorporate with care givers as well as COPD physician and experts with other parts (rehab or psychology.)
Response:
We thank the reviewer for this comment. The authors implemented a figure (Figure 3) with text demonstrating the role of IT solutions. We hope that the provided material is satisfactory for the readers. Please, find enclosed the updated manuscript.

This manuscript is a resubmission of an earlier submission. The following is a list of the peer review reports and author responses from that submission.
Round 1
Reviewer 1 Report
- There is something worthwhile here, but the article is overlong, repetitive and written in an irksome neo-modern psychosocial-babble style.
- The thesis is simple: COPD is frequently complicated by negative social and co-morbidity issues which are currently relatively neglected in a "reactive" model of care. Co-morbidities need to have a prominent place in management strategies and this needs to be systematised so that crucial parts don`t fall through the cracks of the clinical system. This could be said simply and in one to two paragraphs rather than pages.
- The authors pin huge hope on IT systems (ITS) for communication and coordination between therapist(s) and patient, which paradoxically seems to have the danger in itself of atomising care with no-one obviously in charge. It is hinted that the "autonomous" patient will be the driver, but that seems to reflect a particular PC ideology which runs though the document.
- The flaws of ITS as applied to clinical care is hinted at (mixed agenda, e.g. beaurocracy mainly wanting to emphasise monitoring of costs), but issues such as cost, privacy legislation, patient "right" to edit and exclude, lack of buy-in by both patient and medical profession, with the latter due to lack of control but lots of responsibility, and lack of payment for this difficult job, are ignored but need discussion.
- The paper talks a lot around comorbidities but without saying much about what the common ones can be predicted to be. The main one related to COPD per-se is anxiety/depression, due either to having chronic illness or indeed as a direct consequence of nicotine dependance, but we know that this greatly affects patients ability to collaborate with self-management. Long term smoking also of course is related to CV disease; and if the patient is obese then metabolic syndrome, OSA, and again CV disease are highly prevalent. Having a list of common problems that need delineating in each patient would is the basis of a investigational and care-pathway. Why can this not be managed in straight-forward primary care, perhaps backed up be nurse-run community clinics for coordination of the questionnaires and tests (ECG, ECHO, cardiac CT, bloods etc ) needed to define these conditions, but with the team lead by the patient`s GP or internist? This is done well for multi-faceted diabetes care, or example, but a main barrier to application in COPD is lack of funding to reimburse and incentivise such a "care-bundle", though clinicians my need to press harder for this. ITS could certainly facilitate this but is not the main player. In my country one can get paid allied-health involvement, but access is very limited.
- I tend to wince when I see in an introduction COPD being defined primarily as an "inflammatory" airway/lung disease. There is certainly innate immune activation in the airway lumen in smokers and stable COPD, but the key characteristic of the latter is a hypo-cellular and fibrotic airway wall, probably being driven by a gene-reprogrammed airway epithelium (likely also to be the pathogenesis of the high incidence of lung cancer which is also part of the COPD diathesis).